# Experimental and Numerical Study on Microstructure and Mechanical Properties of Ti-6Al-4V/Al-1060 Explosive Welding

**Yasir Mahmood, Kaida Dai \***, **Pengwan Chen \*, Qiang Zhou, Ashfaq Ahmad Bhatti and Ali Arab**

State Key Laboratory of Explosion Science and Technology, Beijing Institute of Technology, Beijing 100081, China; 3820160003@bit.edu.cn (Y.M.); zqpcgm@gmail.com (Q.Z.); ashfaqb@live.com (A.A.B.); arabali83@yahoo.com (A.A.)

**\*** Correspondence: daikaida@bit.edu.cn (K.D.); pwchen@bit.edu.cn (P.C.); Tel.: +86-010-6891-8740 (K.D. & P.C.)

**Abstract:** The aim of this paper is to study the microstructure and mechanical properties of the Ti6Al4V/Al-1060 plate by explosive welding before and after heat treatment. The welded interface is smooth and straight without any jet trapping. The disturbances near the interface, circular and random pores of Al-1060, and beta phase grains of Ti6Al4V have been observed by Scanning electron microscopy (SEM). Heat treatment reduces pores significantly and generates a titanium-island-like morphology. Energy dispersive spectroscopy (EDS) analysis results show that the maximum portion of the interfacial zone existed in the aluminium side, which is composed of three intermetallic phases: TiAl, TiAl$_2$ and TiAl$_3$. Heat treatment resulted in the enlargement of the interfacial zone and conversion of intermentallic phases. Tensile test, shear test, bending test and hardness test were performed to examine the mechanical properties including welding joint qualities. The results of mechanical tests show that the tensile strength and welding joint strength of the interfacial region are larger than one of its constituent material (Al-1060), the microhardness near the interface is maximum. Besides, tensile strength, shear strength and microhardness of heat treated samples are smaller than unheat treated. Smooth particle hydrodynamic (SPH) method is used to simulate the transient behaviour of both materials at the interface. Transient pressure, plastic deformation and temperature on the flyer and base side during the welding process were obtained and analyzed. Furthermore, the numerical simulation identified that almost straight bonding structure is formed on the interface, which is in agreement with experimental observation.

**Keywords:** explosive welding; Ti6Al4V/Al-1060; microstructure; mechanical properties; smooth particle hydrodynamic (SPH)

## 1. Introduction

Welding is a useful technique for joining two similar or different materials. Nowadays, various welding techniques i.e., gas welding, arc welding, laser welding, friction welding, explosive welding etc. have been widely used in many industries. The explosive welding is a solid state welding process in which a flyer plate is accelerated by an explosive and welded with other material in a short interval of time. During the welding process, a high velocity jet removes the impurities on the material surfaces [1]. Explosive welding was used in many mechanical related industries, especially in power plants and aerospace industry. Moreover, it was successfully applied to produce biocompatible materials in the medical related fields [2,3]. This method is advantageous to weld different kinds of metals and alloys that cannot easily be welded by some other means of welding. Materials having excellent mechanical properties (i.e., corrosive resistance, high strength to density ratio, good conductance and

heat resistance, etc.) can be welded with materials having low mechanical properties. This type of welding makes the materials more reliable and cost effective.

Both aluminium and titanium are widely used in the transportation and aerospace industries. Al-1060 belongs to commercially pure aluminium. It is highly malleable and corrosion resistant. However, it is very reactive with air at high temperature and has low mechanical strength. Titanium and its alloys have the highest strength to mass ratio in all naturally existing materials. So their combination can be used as an excellent industrial application. Explosive welding is one of the best choice for bonding these two metal alloys because the two materials have a big difference in their melting points, and mechanical strengths [4]. Furthermore, this combination has been received special attention due to the formation of titanium aluminides at the interface. Titanium aluminides have low density, good oxidation ignition resistance and excellent mechanical properties at high temperatures. Due to these features titanium aluminides get special attention and researchers practised different techniques to develop titanium aluminides. Previously, Perusko et al. [5] investigated the intermetallic formation during Al/Ti welding by using $Ar^+$ ion implementation. Adeli et al. [6] used induction preheating process to synthesis the Ti/Al powder and prepared TiAl phase. Arakawa et al. [7] formed titanium aluminide foam with the help of self-propagating high-temperature synthesis (SHS). They created a foam which contains 60% to 70% porosity due to $TiAl_3$. Titanium aluminides percentage is increased by using diffusion and combustion [8]. DC magnetron sputtering method was opted by Ramos et al. [9] to prepare $\gamma$-TiAl. Furthermore, they controlled activation energy by silver foil. Romankov et al. [10] prepared $TiAl_3$ intermetallic by using thermal deposition technique.

Kahraman et al. [11] studied the complex microstructure of explosively welded Ti6Al4V and aluminium alloy using different charge to mass ratios. They found that hardness and corrosion were increased at the interface with the increase of the charge to mass ratios. E et al. [12] evaluated the tensile properties of the explosively welded Ti/Al sample with a load applied along parallel and perpendicular directions to the interface. Xia et al. [13] observed the micro grains and recrystallisation of titanium alloy in the interface. Fronczek et al. [14] and Bazarnik et al. [15] studied mechanical properties and microstructure of the explosively welded titanium alloy with aluminium alloy.

Additionally, the Ti/Al interface study is essential to check the quality of welded joints. According to Ege et al. [16], at the Ti/Al interface, the aluminium percentage content reinforces the Ti/Al joints, providing stability and increasing the strength up to 825 MPa. Inal et al. [17] found that the straight interface is more suitable than the wavy interface because it accompanied by excessive heat generation and could produce weak and brittle intermetallics. Ege et al. [18] presented that the heat treatment of the Ti/Al welded sample did not affect the stability, therefore could be used for high temperature environment. Multilayered combination for explosive welding of titanium and aluminium is one of the best options to fabricate titanium aluminides. Lazurenko et al. [19] welded forty layers and Mali et al. [20] joined 23 of Ti-Al by using explosive welding technique. They produced $TiAl_3$ stable intermetallics by using sintering at 640 °C temperature and 3MPa pressure. Furthermore, Bataev et al. [21] studied the thickness of $TiAl_3$ from top to bottom plates and compared the results with post heat treated samples. Likewise, Foaden et al. [22] analysed the $TiAl_3$ formation process after annealing of the explosively welded composite plate. Fan at al. [23] prepared multilayered TiAl foils by using vacuum hot pressing and studied the foil thickness effect. Similarly, Fan et al. [24] welded 5 Ti-Al plates by using an explosive welding technique to analyse the microstructure and mechanical properties of composite plates.

Simulation is an excellent tool to predict the explosive welding conditions, i.e., impact velocity, pressure, plastic strain, etc. Many authors applied various approaches to simulate explosive welding phenomenon. Recently, Mousavi et al. [25] used ANSYS AUTODYN to simulate the straight, wavy, jetting morphology and humps formation during the welding process. Nassiri et al. [26] replicated the jetting phenomena with the help of SPH method, and exercised Arbitrary Lagrangian-Eulerian (ALE) method to obtain other welding parameters, i.e., interface shape, temperature, the velocity of material, shear stress. Wang et al. [27] imitated the shear stress and effective plastic strain of materials by using

the SPH approach. Wang et al. [28] reproduced the whole explosive welding phenomena with the help of material point method (MPM) in C++ program.

In this paper, microstructure and mechanical properties of the welded composite plate (Ti6Al4V and Al-1060) were investigated. The SPH approach with the help of LS-DYNA was used to understand the welding conditions i.e., interface morphology, transient pressure, temperature and plastic deformation.

## 2. Experimental Procedure

### 2.1. Materials and Explosive Welding Setup

Ti6Al4V was used as a flyer plate and Al-1060 as a base plate because titanium had lower thermal diffusivity than aluminium [29]. Both plates with the size of 200 mm × 150 mm × 3 mm were welded in air. The schematic experimental setup is shown in Figure 1. Both plates were arranged in parallel configuration with a standoff distance of 5 mm. Ammonium nitrate and fuel oil (ANFO) with 30 mm thickness (packing density is 670 Kg/m$^3$ and detonation velocity is 2600 m/s) was used to accelerate the flyer plate. The sand was employed as an anvil. The welded samples were heat treated at 525 °C for 4 h and then cooled in open air. Inal et al. [17] reported that the bond strength remains constant at this heating temperature and time duration.

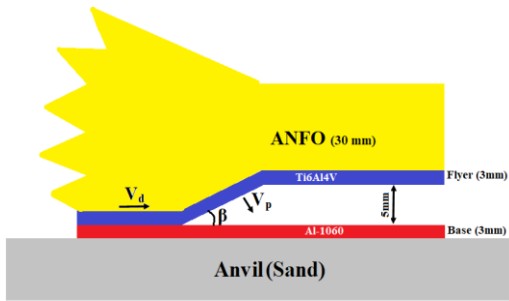

**Figure 1.** Schematic diagram of explosive welding.

### 2.2. Microstructure Test

For microstructure analysis, samples were cut from the middle of the welded plate approximately 100 mm away from the detonating point. The samples were mounted in an epoxy such that the welding interface was parallel to the detonation direction. The samples were cleaned with sandpapers (Grit 320, 1000, 1500, 2000), and then polished with ceramic powder. Finally, the samples were soaked in the Kroll reagent (distilled water 92 mL, nitric acid 6 mL, hydrofluoric acid 2 mL) for 20 s to examine the apparent microstructure.

The microstructure and elements distribution at the interface was observed by using scanning electron microscope (SEM, Hitachi S-4800, Tokyo, Japan) at an acceleration voltage of 15 kV. SEM elemental scan images were further edited to combine both elements (Ti-Al) using CoralDraw (Coral Corporation, Ottawa, ON, Canada).

### 2.3. Mechanical Test

The Vickers hardness tests were conducted along the perpendicular direction of the interface. The hardness tests were carried out on a microhardness machine using a 0.98 N load for 15 s.

For mechanical tests, stratified samples were obtained from the welded plate before and after heat treatment. Tensile test and shear test samples (as seen in Figure 2a,b) were performed on material testing machine (MTS) with a loading rate of 5 mm/min and 1.8 mm/min respectively to measure the tensile properties of the welded portion. For three point bending test, the sample with a thickness of 5 mm and a length of 100 mm was conducted along the perpendicular direction of the flyer-base interface.

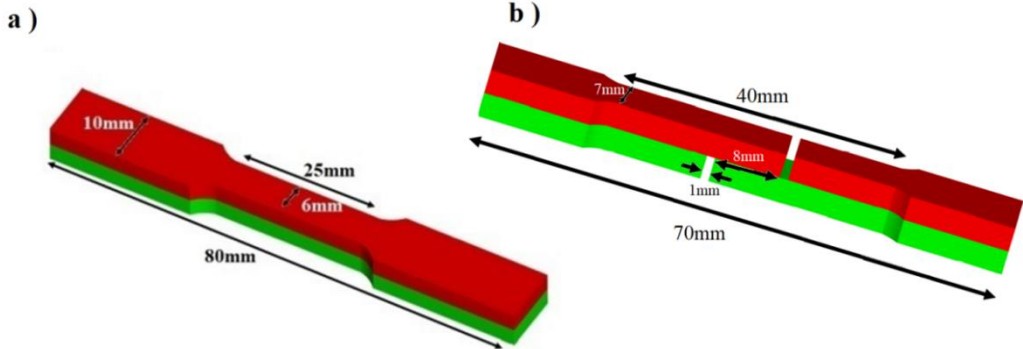

**Figure 2.** Schematic diagram for (**a**) tensile test (**b**) shear tensile test.

## 3. Simulation of Explosive Welding

ANFO has non-ideal detonation behavior [30]. It is difficult to simulate it with Jones-Wilkins-Lee (JWL) equation because it has a long reaction zone, so energy and momentum at Von-Neuman Spike cannot be neglected [31]. Therefore empirical formula was used to calculate the plate velocity and impact angle.

### 3.1. Calculation of Plate Velocity

Deribas et al. [32] and Manikandan et al. [33] developed a distance dependent empirical relation to calculate the impact angle between flyer and base plate. The impact angle is defined as,

$$\beta(rad) = \left( \sqrt{\frac{k+1}{k-1}} - 1 \right) \frac{\pi}{2} \frac{R}{\left( R + A + \frac{Bt_e}{S} \right)} \tag{1}$$

where $A = 2.71$, $B = 0.184$, $t_e$ is the thickness of explosive, $S$ is the standoff distance between the flyer and base plate. Polytropic exponent $k$ can be measured with the by Gurney velocity [32].

$$k = \sqrt{\frac{D^2}{V_g^2} + 1} \tag{2}$$

where $V_g$ represents the Gurney velocity, which can be calculated by using the following formula [34–36].

$$\sqrt{2E}(m/s) = 600 + 0.52\frac{D}{\sqrt{\gamma + 1}} \tag{3}$$

where $\gamma$ is a ratio of specific heat constant. For ANFO, the value of $\gamma$ is 2.881 [37]. Plate velocity can be calculated by Crossland relation [38].

$$V_p = 2Dsin\left(\frac{\beta}{2}\right) \tag{4}$$

Using the above equations for the current experimental conditions, plate velocity $V_p$ with 5 mm standoff distance is 707 m/s and the impact angle $\beta$ is 15.04°.

### 3.2. Equation of State and Constitutive Model

Mei Gruneisen equation of state is mostly used for shock wave propagation [27]. This equation gives us a relation between pressure and volume under shock conditions at a given temperature. Johnson-Cook material model was used for the simulation of explosive welding, because it can

successfully predict the high deformation and Von-Misses yield stress of the material. Johnson-Cook equation can be written as

$$\sigma = (A + B\varepsilon^n)\left(1 + C \ln\dot{\varepsilon}_p\right)(1 - T^{*m}) \tag{5}$$

where $\varepsilon$ is plastic strain, $C$ is strain rate constant, $\dot{\varepsilon}_p$ is plastic strain rate, $n$ is hardening exponent, $A$ is yield strength of the material, $m$ is softening exponent, $B$ is strain hardening coefficient and $T^*$ is homologous temperature and equal to $T^* = \frac{(T - T_{room})}{(T_{melt} - T_{room})}$. The parameters of material models for Ti6Al4V and Al-1060 are listed in Table 1.

**Table 1.** Parameters of material model and equation of state.

| Material | A (MPa) | B (MPa) | N | C | M | Density (Kg/m³) | $C_o$ (m/s) | G | S |
|---|---|---|---|---|---|---|---|---|---|
| Ti-6Al-4V [39] | 1098 | 1092 | 0.93 | 0.014 | 1.1 | 4430 | 5130 | 1.23 | 1.028 |
| Al-1060 [40] | 66.56 | 108.8 | 0.23 | 0.029 | 1 | 2707 | 5386 | 1.97 | 1.339 |

### 3.3. Smooth Particle Hydrodynamics (SPH)

Numerical gridless lagrangian hydrodynamics simulation was carried out with the SPH method in ANSYS/LS-DYNA (Ansys 14.5, LSTC, Livermore, CA, USA). In SPH method, Kernel approximation of field variable at given points is applied to simplify the conservation equations. Discrete particle helps to find out field information while neighboring particles are used to solve the integrals. If neighboring particles are expressed by subscript $j$, then field variable for non zero Kernel approximation can be described as follow:

$$f(r) \cong \sum \frac{m_p}{\rho_p} f_j W(|r - r_j|, h|X|, h) \tag{6}$$

where $m_p$, $\rho_p$ are particle mass and density respectively, $f(r)$ is a field variable, $r$ represents the location of particle, $h$ shows the length at which particle is effected by the neighboring particles. For the current simulation of the explosive welding, it is simplified with the collision of the flyer and base plate at a certain velocity and angle. These two parameters are calculated by Equation (1) and Equation (4), respectively. The schematic model for SPH simulation is shown in Figure 3, with a particle size of 0.5 µm.

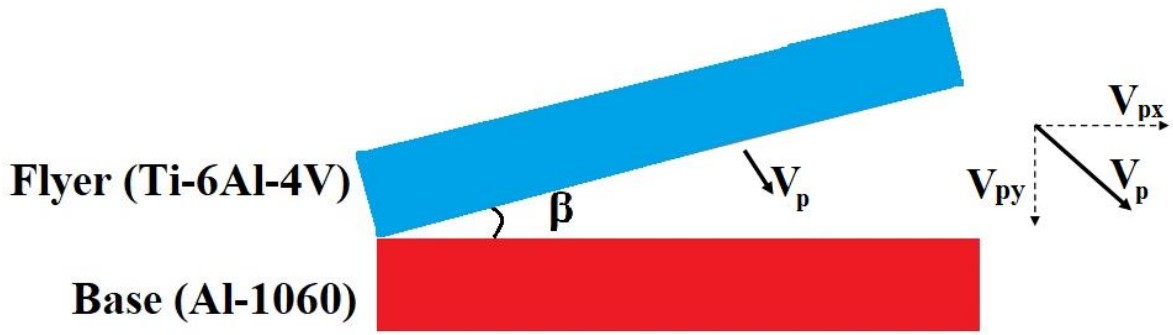

**Figure 3.** Model for SPH simulation.

## 4. Results and Discussion

### 4.1. Microstructure of Welding Interface before Heat Treatment

The SEM results of the welded sample before heat treatment are displayed in Figure 4. It shows that welding interface is smooth and flat without delamination. This type of smooth Ti/Al interface joint pattern was previously observed by Bazarnik et al. [15] for Ti6Al4V/Al2519 welding and Ege et al. [16] for Ti6Al4V/Al6061 multilayer welding.

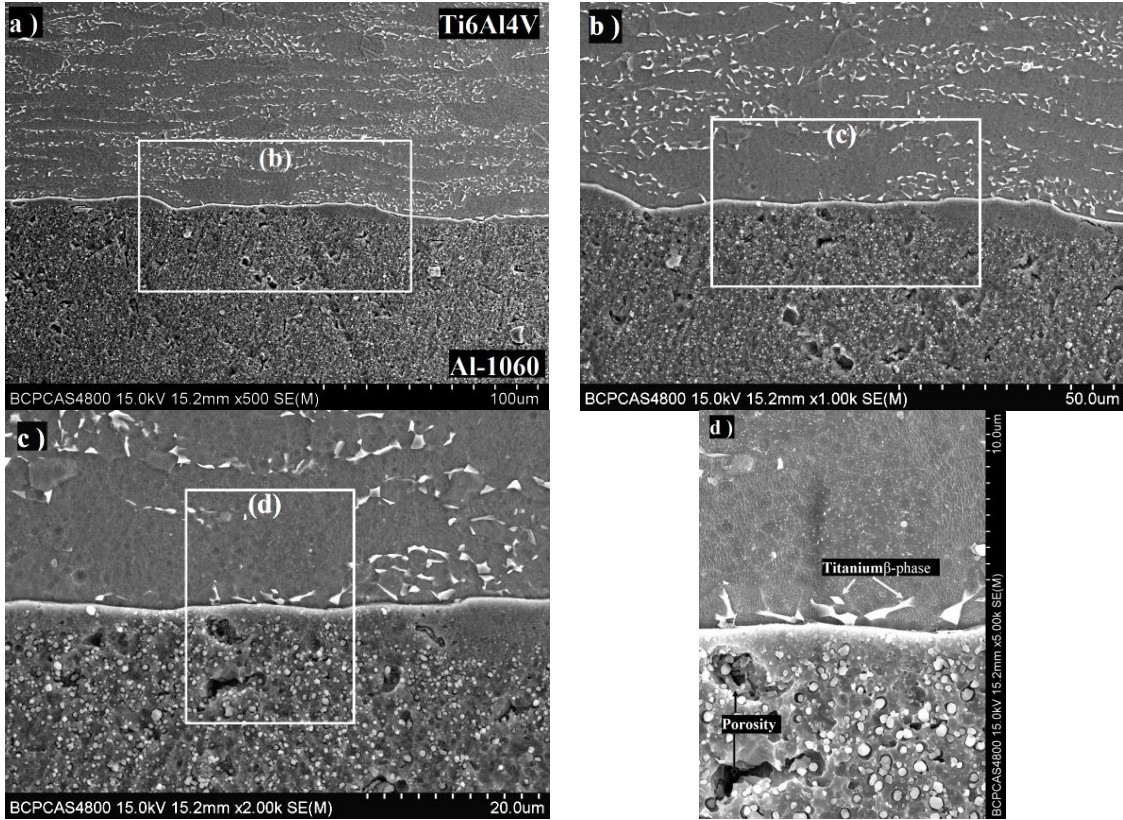

**Figure 4.** SEM image of explosively weldedTi6Al4V/Al1060 interface before heat treatment at magnifications of (**a**) 500×, (**b**) 1000×, (**c**) 2000×, (**d**) 5000×.

Seeing from Figure 4a–d, both metals have some disturbances in microstructures near the interface, especially aluminium side shows the maximum disturbances. High pressure and deformation during explosive welding process may generate recrystallization of microstructure. These phenomena near the interface formed under high pressure were previously observed by Zhang et al. [41] and Gloc et al. [42]. Furthermore, Figure 4d shows that Al-1060 contains some circular and random pores. These pores are formed during severe deformation and high temperature gradient, causing rupture of local surfaces. Raoelison et al. [43] analyzed the formation of pores during the welding process and stated that these pores might be the origin of the crack propagation. The same pores were observed by Su et al. [44] during the welding process of Fe-Al. As shown in Figure 4d, Ti6Al4V microstructure near the interface is also altered, especially beta phase grains that are elongated toward the detonation direction. During the explosive welding process, due to high strain and pressure, the temperature of the interfacial zone is abruptly raised. This temperature increment is high enough to change the phase of the plate [25]. Therefore, Ti6Al4V is most likely to be converted to the beta phase during the collision. However, the cooling rate is very high in this process, the titanium phase change process only takes a short interval and quickly turns back to the alpha phase [38]. According to Tomashchuk et al. [45] the titanium beta phase is more likely to react with aluminium than alpha phase to form titanium aluminide. The intense plastic deformation and elongation were also observed by Murr et al. [46] and Kacar et al. [47].

The interface between titanium and aluminiumhas a very complicated structure, so EDS technique is used to understand the microstructure. Figure 5a shows an elemental scan near the interface region, indicating that the Ti6Al4V/Al-1060 interface is smooth and flat. Figure 5b reveals that aluminium element counts in the base plate start decreasing about 0.86 μm away from the interface. This decrease in aluminium counts continues in flyer plate and gets normal counts after 0.24 μm thickness from the interface. So the total width of the interfacial zone is 1.1 μm with maximum portion existing in the

base side (aluminium). Slope difference shows that aluminium counts near interface decline faster in the base plate as compared to the flyer plate side.

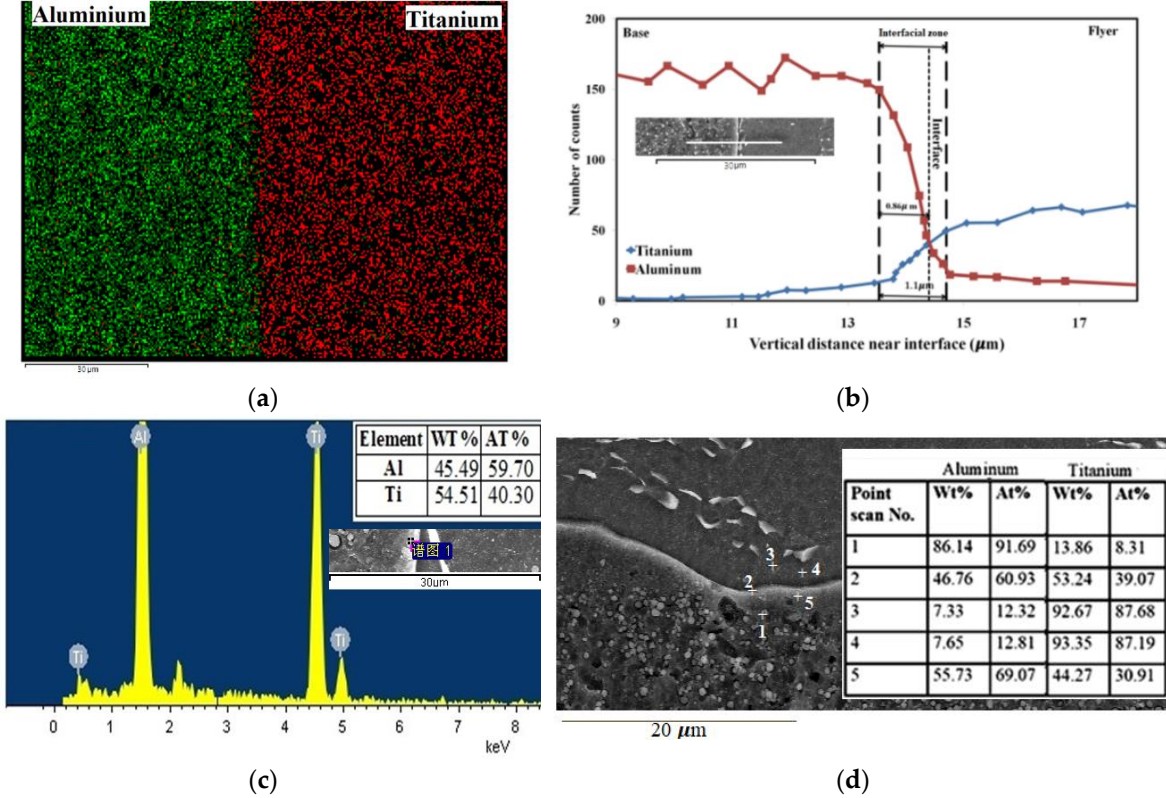

**Figure 5.** (**a**) Elemental scan of aluminium and titanium elements along with welding interface, (**b**) graphical representation of linear element analysis at the interface, (**c**) EDS point spectrum at the interface, (**d**) Ti-Al atomic distribution near interface at selected points.

Parasithi et al. [48] reported that materials with higher thermal conductivity keep a maximum percentage in interfacial area. Since aluminium has almost more than 25 times higher thermal conductivity as compared to titanium, that is why maximum microstructural changes are observed in the base plate. Manikandan et al. [33] gave another demonstration about the interfacial zone. According to their study, at the interface, the atomic concentration of base material is relatively higher than the flyer material.

Due to the extreme conditions in explosive welding process, chemical equilibrium was not achieved. Consequently, this may exhibit various kinds of intermetallics at different metastable equilibrium states [49]. Ti-Al has three main titanium aluminides TiAl, $TiAl_3$, and $Ti_3Al$. $TiAl_2$ is another equilibrium phase of titanium aluminide. However, this phase is formed under the overlap condition with the TiAl phase [14]. Figure 5c shows that the atomic percentage of aluminium in the interface is 60% and titanium is about 40%. It indicates that this region is formed by overlapping of two different phases of Ti-Al, i.e., TiAl + $TiAl_2$. Fronczek et al. [14] observed this phase with the help of X-ray synchrotron reflection method. Similarly, in Figure 5d, EDS point scan exhibits that point 2 has a chance of overlapped phase TiAl + $TiAl_2$ and point 5 may contains $TiAl_3$ phase. While the remaining points show no external interference to other elements.

### 4.2. Microstructure of Welding Interface after Heat Treatment

Figure 6a–d with different selected area magnifications show that the interfacial zone becomes wider after 4 h heat treatment. Pores are significantly decreased and grains are rearranged. High deformation can produce pores and cracks during explosive welding. Consequently, when the stress of the composite plate is relieved by heat treatment process, then these cracks can generate the island/peninsula-like morphology. Furthermore, the formation of this island/peninsula pattern is influenced by the detonation force and metal vortex flow. Previously, many researchers observed this type of morphology [16,48,49]. Figure 6d shows this titanium-island-like shape (marked by red line) in the aluminium rich area. These types of islands may affect the mechanical performance of the welded plate [50,51].

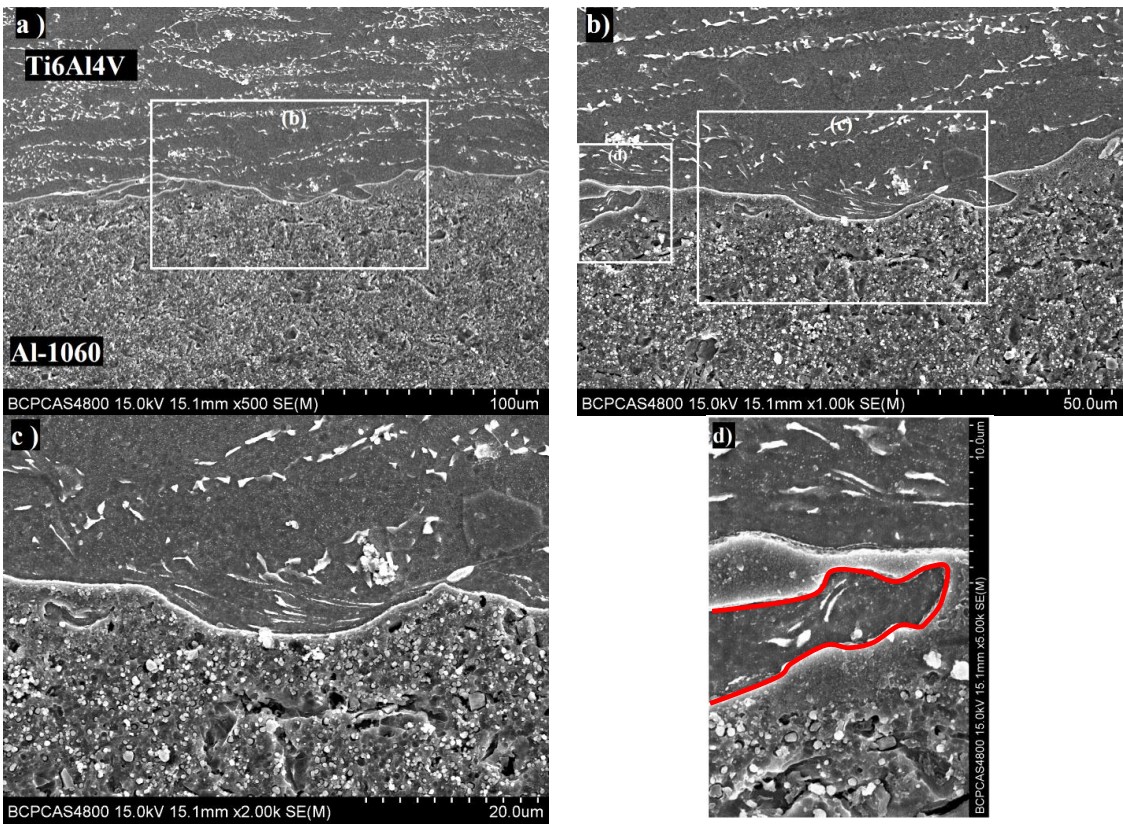

**Figure 6.** SEM images of explosively welded Ti6Al4V/Al1060 interfaces after heat treatment at different magnifications of selected regions (**a**) 500×, (**b**) 1000×, (**c**) 2000×, (**d**) 5000×.

The elemental scan of Ti-Al interface (Figure 7a) shows that at the interface, an island (as indicated by arrow) is formed with titanium as the dominant element while the surrounding area consists of titanium aluminides. Their detailed atomic percentage can be determined by using the EDS line and point scans (Figure 7b–d). As shown in Figure 7b, two positions are selected for line scan EDS. One line passes through islands and the neighboring area of interface (Figure 7c), while the second line passes through interface and surroundings (Figure 7d).

Figure 7b describes the point elemental scan near interface and island morphology. The point scans 3 and 5 show that the area near the interface contains titanium aluminides. Based on EDS atomic percentage analysis, there is a possibility of the existence of most stable titanium aluminide (TiAl$_3$). It can further be verified from line scan in Figure 7c, which shows that region 2 has higher titanium counts as compared to the base plate side. It indicates that the post heat treatment process influences the aluminides equilibrium state. Points 1, 2 and 4 scans show that there is no reasonable

external involvement of any element in each side. Particularly point 4 is situated in the mid of island morphology. This point shows that the island is formed from the flyer plate.

EDS line scan (Figure 7c) shows that the interfacial zone is expended up to approximately 11.55 µm, which is almost 10 times larger than unheated sample (Figure 5b). In this interfacial area, the titanium island surrounded by Ti-Al intermetallics is observed, which has a width of 5 µm. Titanium aluminides of different equilibrium phases have been observed in the area between island and interface, which is expanded up to 4.6 µm. Heat treatment reduces the number of element counts in the unit area, increases the titanium penetration in aluminium and stretched the interfacial zone. Furthermore, the equilibrium phases of titanium aluminide are formed. Especially, $TiAl_3$ is dominated as compared to all other phases. This kind of variations were also observed by Gloc et al. [42] and Fronczek et al. [14].

Figure 7d shows that interfacial zone becomes more extensive than the sample before heat treatment. Although titanium is penetrated into aluminium up to 3 µm, the high level of Ti-Al mixing is in the range of 1.7 µm with a maximum area existed in the aluminium side. Comparison of the Figure 7c–d indicates that the interfacial area has a difference of about 9.8 µm. It shows that the post heat treatment process makes the interface irregular shape. Additionally, it is noticed that titanium aluminides have existed in the area between the titanium island and the flyer plate. This area did not directly expose during the collision of the plates. It implies that titanium aluminides was created during the heat treatment process.

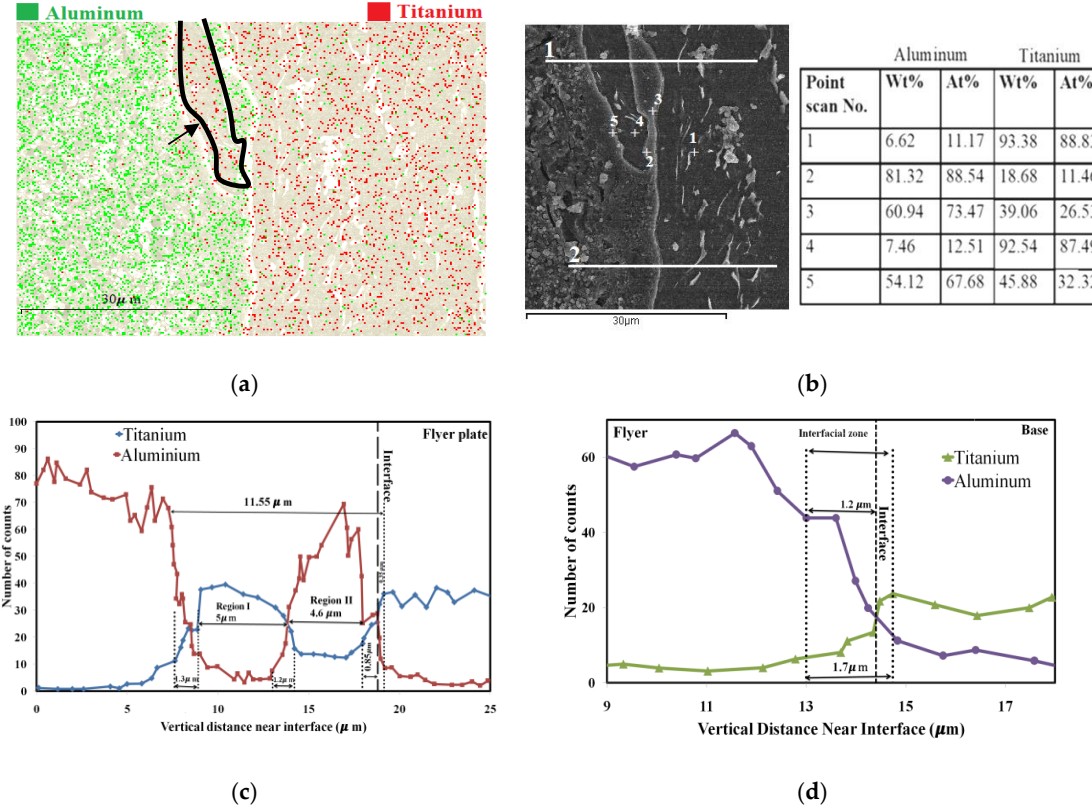

**Figure 7.** Pictures and EDS after heat treatment (**a**) Ti-Al mapping near interface, (**b**) Ti-Al atomic distribution near interface at selected points, (**c**) linear distribution of Ti-Al near interface (at position 1 in Figure 7b), (**d**) Linear distribution of Ti-Al near interface (at position 2 in Figure 7b).

## 4.3. Mechanical Tests

Tensile tests were conducted to study the mechanical response of the welded material. Two samples were prepared from normal and heat treated welded plates, as shown in Figure 8.

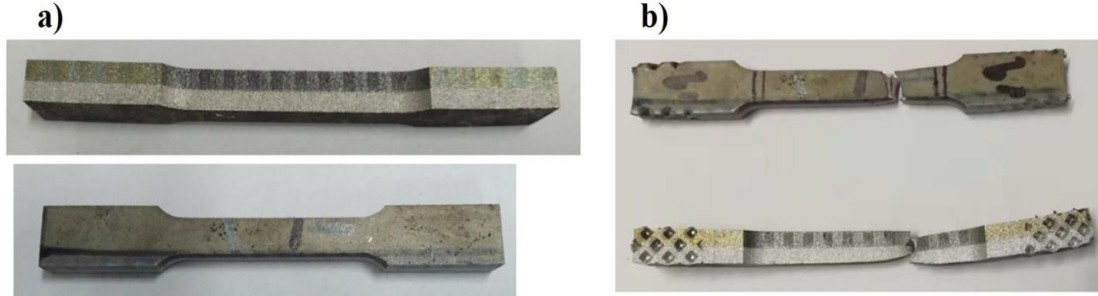

**Figure 8.** Tensile test samples (**a**) before the test; (**b**) after the test.

For successful bonding, the tensile strength of the welded sample should be higher than the weaker material used for welding [50]. Ege et al. [16] proposed that the yield strength of welded material depends on aluminium volumetric percentage and interfacial density. Figure 9 shows the stress-strain curves of the tested samples before and after heat treatment. After heat treatment, the tensile strength of the welded sample was decreased, while the elongation and plasticity were increased. Furthermore, Figure 9 shows that a small jerk occurred at the end of elastic limit of the unheated sample. This jerk state indicates that failure begins from the weak material (Al-1060) before the tensile strength is reached. The experimental results of the tensile tests were compared with the mechanical properties of the Al 1060 [51] and Ti6Al4V [52] based on their percentage thickness in the welded plate (Table 2). This approximation helps us to estimate the tensile properties of the welded sample [16,53]. Detailed results are shown in Table 2.

It is indicated that both samples have better results than the calculated values. Furthermore, experimental results exhibit that the tensile strength and yield stress of heat treated samples are reduced as compared to the unheated sample, but elongation is improved from 20% to 23%.

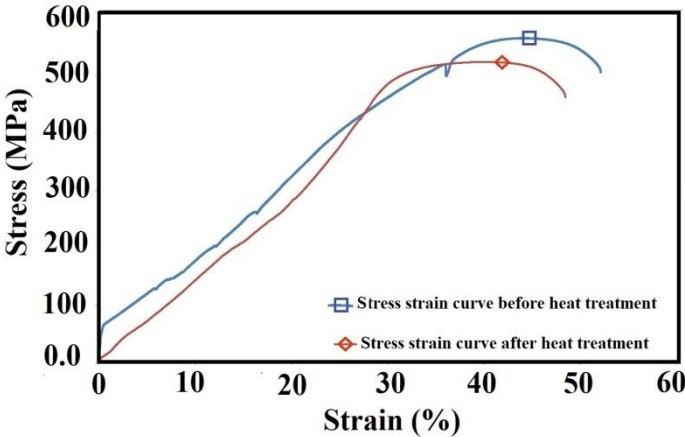

**Figure 9.** Engineering stress-strain profile before and after heat treatment.

**Table 2.** Tensile test results of the welded samples together with specific materials.

| Samples | UTS (MPa) | Yield Stress (MPa) | Elongation (%) |
|---|---|---|---|
| Ti-6Al-4V [52] | 947 | 872 | 13 |
| Al-1060 [51] | 97 | 110 | 28 |
| Calculated value | 509 | 446 | 20.7 |
| Before heat treatment | 560 ± 4 | 486 ± 8 | 20 ± 1 |
| After heat treatment | 525 ± 1 | 462.5 ± 8 | 23 ± 2 |

Three points bending test results show that no fractures or cracks were observed in any samples for both before and after heat treatment, as seen in Figure 10a,b.

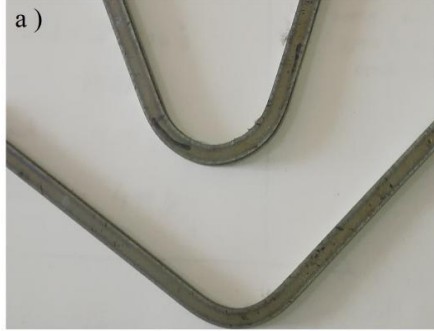
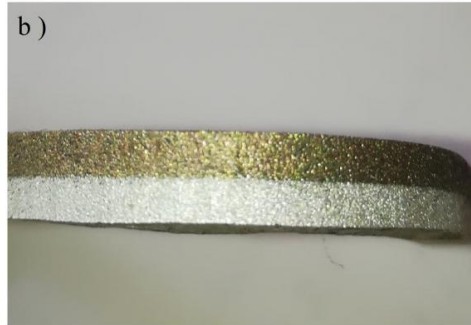

**Figure 10.** Three points bending test (**a**) side view of 90° and 180° bent sample; (**b**) bending interface.

In addition to the bending test and tensile test, the flat shear test was conducted to verify the welding joint strength for both samples before and after heat treatment. Figure 11a indicates that the tensile strength of aluminium decreases after the heat treatment. Figure 11b shows that the deformation starts in the aluminium and the joint displays no disturbance. This result reveals that the joint is stronger than the Al-1060 tensile strength ~40MPa.

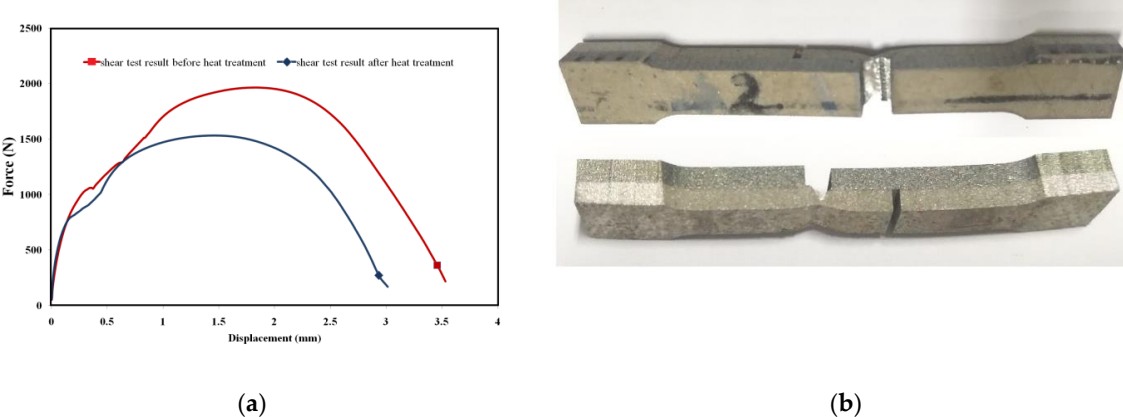

(**a**)　　　　　　　　　　　　　　　　　　　　　　　　　　　　(**b**)

**Figure 11.** (**a**) Shear force versus displacement (**b**) fracture morphology after shear test.

Figure 12 shows that near the interface, the microhardness is maximum. Its value is about ~412, higher than the flyer and base standard values (flyer plate ~350, base plate ~30), which is due to high value of heat produced during explosive welding caused annealing at the contact point [47]. Heat treatment relaxed the interface and widened the interfacial zone, which affected the hardness value and caused it to decrease to ~390 at the interface. As moving away from the interface, hardness becomes approaching to the typical values of flyer and base plate

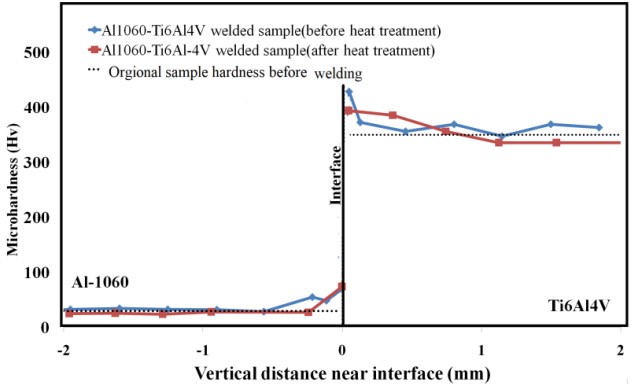

**Figure 12.** Microhardness (Hv) profiles before and after heat treatment.

### 4.4. Simulation Results

Figure 13a shows that smooth interface obtained by numerical simulation, which is in good agreement with the experimental results (see in Figure 4). Since Ti6Al4V is a very hard material and Al-1060 is soft, the flyer can easily penetrate into the base material. Furthermore, Figure 13a explains that some partial wavy shapes patterns appear, but due to high impact velocity and density difference, these waves are suppressed and became almost flat. Figure 13b exhibits that jet is formed during this process and the maximum portion of the jet consisted of base plate (Al-1060).

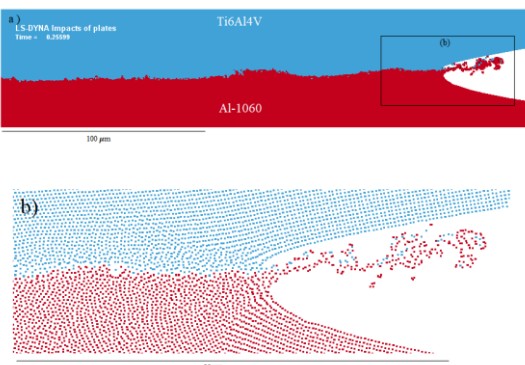

**Figure 13.** Simulation results for (**a**) Ti6Al4V/Al-1060 interface and (**b**) jet formation.

Figure 14a,b show that near the contact point, the pressure distribution is behind the jet. It means that the collision velocity is subsonic, which is an essential requirement for the bonding of materials. According to Blazynski et al. [4], the bonding contact pressure should be higher than material yield strength. Pressure contour plots (Figure 14a,b) illustrate that the transient pressure profile at all contact points during the collision is more than 10 GPa. Figure 14c supports that, at time 0.256 µs, the peak flyer transient pressure rises to 13.1 GPa, which is considerably higher than the yield strength of the flyer and base plate.

Additionally, it was observed (Figure 14c) that in the base plate, the pressure and impulse were slightly lower than that of the flyer. According to Holtzman et al. [54] investigation, if the base plate had a greater contribution to the jetting, there would be more transient pressure in the flyer side. Similarly, Mousavi et al. [37] simulated the same pressure difference between the flyer and base plates.

Furthermore, Figure 14c shows that there is a slight hump in the pressure-time curves before reaching the peak value. This hump indicates that the deformity had begun before the arrival of the shock wave. Cowen et al. [55] reported that deformation and jet formation were necessary for material bonding.

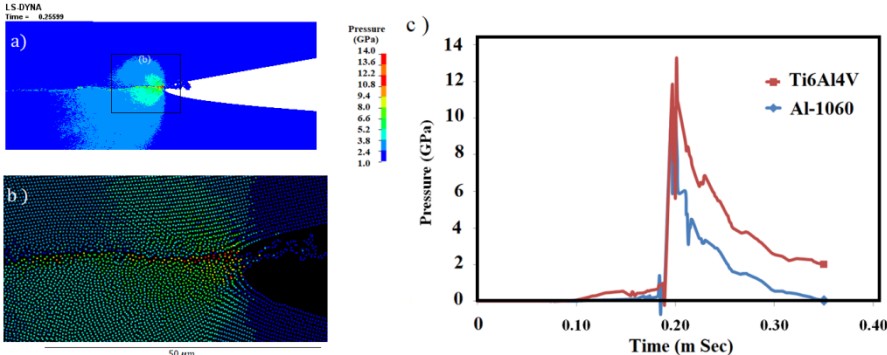

**Figure 14.** Numerica Simulation results for (**a**,**b**) pressure contour and (**c**) pressure-time graph at the interface.

The high impact creates shearing at the interface that causes to rise in heating and produce a bond between different metals. Kinetic energy during impact is converted into plastic work that causes the rise of temperature. Plastic deformation is not possible to witness experimentally, so simulation is an excellent way to explain the process.

Simulation results show that along with interface (Figure 15a–d), the maximum plastic strain is increased up to 7, while averagely its value is more than 5 throughout the interface. This strain value is enough to verify that at this impact velocity, the welding should be possible (the minimum plastic deformation required for welding should be more than 0.25 (Ti-MS) [37]). Figure 15a,b show that high plastic deformation expands up to 2 μm near the interface, which is similar to the experimentally measured interfacial area. It indicates that maximum deformation of the region generates the interfacial zone.

Figure 15c explains that this deformation is purely localized. As we move away from the interface, this deformation abruptly decreases to the minimum level. Furthermore, after 10 μm the flyer shows no plastic deformation, while the base plate has accommodated this deformation up to 30 μm.

Figure 15d illustrates that at the same impact point, the flyer and base plates have different values of plastic strain. The base plate has a plastic strain value almost double than that of the flyer plate. It is just because of the differences in their density and mechanical properties.

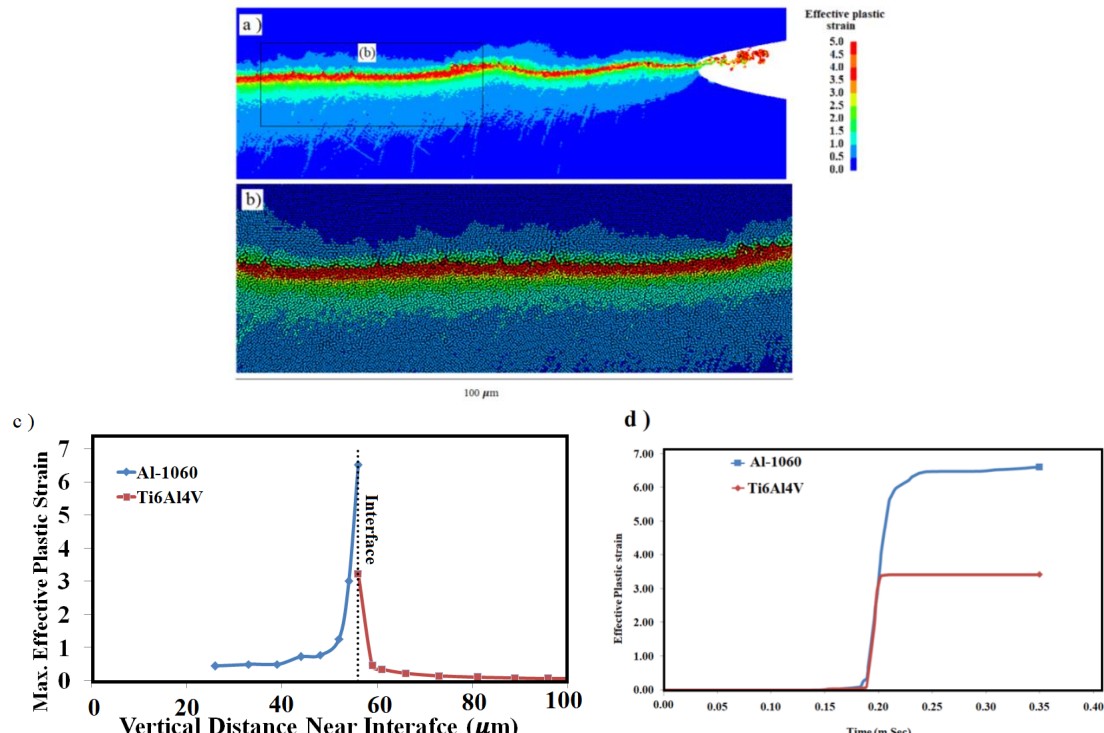

**Figure 15.** (**a**,**b**) Distribution of plastic deformation contour along with the interface, (**c**) graphical representation of plastic strain along the vertical distance of interface, and (**d**) plastic strain plot with respect to time at the interface.

During explosive welding, high pressure and plastic strain raise the temperature locally and abruptly because this phenomenon occurs in a very short interval of time and the cooling rate during this process is very high about $10^5$–$10^7$ K/s [38].

In contour plots of temperature (Figure 16a–c), it is indicated that the temperature increment is enough to melt for both metals. However, both the temperature increments of flyer and base plates are not the same. At the flyer plate side, the average maximum temperature is 3000 K, while the base has an average of 1200 K. It is also observed that along with the interface, the peak temperatures have no fixed value. Figure 16d shows that the flyer has attained its melting point temperature in a

very short time during the impact and then cools down immediately. While at the base plate side, the melting temperature sustains longer as compared to flyer. Obtained the melting point implies that the interfaces of both materials are more likely to form intermetallics. Furthermore, the interfacial zone behaves like the fluid flow and gets elongated toward detonation direction. Additionally, this process causes the refinement of grains near the interface.

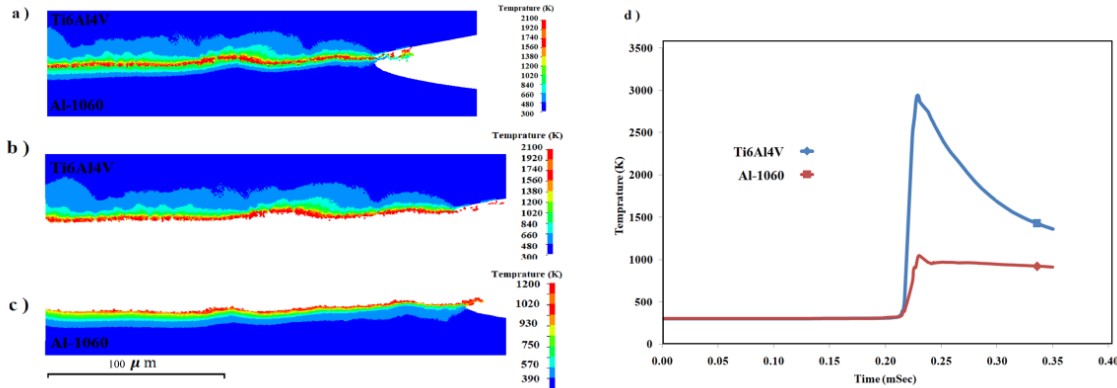

**Figure 16.** Plot of the temperature distribution at (**a**) the interface, (**b**) flyer side, (**c**) base side and (**d**) temperature profile of flyer and base plates at the interface.

## 5. Conclusions

Ti6Al4V was successfully welded with Al-1060 by explosive welding. The welded interface between Ti and Al was smooth and straight without any jet trapping. The maximum portion of the interfacial zone existed in the base side (Al-1060) where different phases of titanium aluminide were observed. Mechanical results, i.e., tensile test, bending test, shear test and Vickers hardness test, showed that welding quality was not highly affected by these titanium aluminides.

Heat treatment process stretched the interfacial zone with some titanium island/peninsula like shape. Due to this, strength of welded material was decreased as compared to the normal welded sample, but ductility was improved.

Numerical simulation depicted that impact pressure at all contact points had larger values than the yield strength of both welded materials, which is one of the basic requirements to meet the welding conditions. Furthermore, simulation results showed that in the interfacial zone, plastic deformation had values more than 5 and both materials obtained their melting points during impact. Melting of both materials provide a reason to form titanium aluminides. Since pressure, plastic deformation and temperature distribution for both materials (flyer and base) had different values, therefore, both materials had different interfacial thickness.

**Author Contributions:** Conceptualization, Y.M. and P.C.; explosive welding, Y.M. and Q.Z.; microstructure examination and mechanical tests, Y.M., A.A.B. and A.A.; numerical simulation, Y.M., K.D. and Q.Z.; writing–original draft manuscript, Y.M.; writing–review and editing, K.D. and P.C.

**Funding:** This research was funded by the National Natural Science Foundation of China (Grant No. 11521062) and State Key Laboratory of Explosion Science and Technology, Beijing Institute of Technology (Grant No. ZDKT18-01).

**Conflicts of Interest:** The authors declare no conflict of interest.

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
