# Peer review of "Experimental and Numerical Study on Microstructure and Mechanical Properties of Ti-6Al-4V/Al-1060 Explosive Welding"

_metals, doi:10.3390/met9111189_

Round 1
Reviewer 1 Report
The article is interesting and useful, well organized and legible. The authors described research on interface structure, microstructure and mechanical properties of laminated composites based on Ti-6Al-4V alloy and Al-1060 aluminum connected by means of explosive welding. Some of the tested samples were subjected to additional heat treatment.
Despite the fact that the idea of such a study is not entirely new, the manuscript contains new results, especially in the field of numerical simulation, during which it has been shown that an almost simple binding structure is formed on the interface, which is consistent with experimental observation. Based on the results obtained, a very detailed description of the microstructure of the connection zone was made, especially in EDS studies.
The analysis presented is logical and the number of references is sufficient. In general, the results of explosive welding simulation and experimental tests included in the article allow to extend the scope of inference. I suggest using this option.
Author Response
1

Reviewer 2 Report
The paper presents original and current results. Results are presented in good form. It is recommended to review the article for English language..Author Response
2

Reviewer 3 Report
In the paper I found some inaccuracies that should be explained and corrected:
General remark: What is Author’s opinion about possibility of using metal forming processes (cold or hot rolling) for Ti6Al4V/Al multi-layer materials? There is a problem in a different mechanical properties Ti compared to Al. There also is problem in cold metal forming of Ti6Al4V alloy. The references used are sufficient for the paper's issues clarification. However each one (two) of the quoted references should be discussed individually and demonstrate their significance to the work. It is not necessary used four or even six references in one bracket: [5-10], [19-24]. page 2, line 70: is Arbitrary Lagrange Euler (ALE) method, should be Arbitrary Lagrangian-Eulerian Method Figure 1: is anvail, should be anvil. Table 1: Authors should add some information about plastometric tests, not only quoted literature. As we know during the explosive welding process, there is a high temperature gradient, from the ambient temperature up to melting of the metal. That’s why during the numerical modelling of the explosive welding process there is a problem with a description of the flow stress. page 6, line 226: is Golc, should be Gloc Fig. 11a is not legible Did a jerk appear during the shear test? What was shear stress value in MPa? Fig. 14: in the description of this Fig. there is no c)In the final evaluation I conclude that the reviewed article is suitable for printing in Metals journal.
Author Response
3
